# Predictive Inference Is Free with the Jackknife+-after-Bootstrap

Byol Kim[*]        Chen Xu[†]        Rina Foygel Barber[‡]

## Abstract

Ensemble learning is widely used in applications to make predictions in complex decision problems—for example, averaging models fitted to a sequence of samples bootstrapped from the available training data. While such methods offer more accurate, stable, and robust predictions and model estimates, much less is known about how to perform valid, assumption-lean inference on the output of these types of procedures. In this paper, we propose the jackknife+-after-bootstrap (J+aB), a procedure for constructing a predictive interval, which uses only the available bootstrapped samples and their corresponding fitted models, and is therefore "free" in terms of the cost of model fitting. The J+aB offers a predictive coverage guarantee that holds with no assumptions on the distribution of the data, the nature of the fitted model, or the way in which the ensemble of models are aggregated—at worst, the failure rate of the predictive interval is inflated by a factor of 2. Our numerical experiments verify the coverage and accuracy of the resulting predictive intervals on real data.

## 1 Introduction

Ensemble learning is a popular technique for enhancing the performance of machine learning algorithms. It is used to capture a complex model space with simple hypotheses which are often significantly easier to learn, or to increase the accuracy of an otherwise unstable procedure [see 14, 27, 29, and references therein].

While ensembling can provide substantially more stable and accurate estimates, relatively little is known about how to perform provably valid inference on the resulting output. Particular challenges arise when the data distribution is unknown, or when the base learner is difficult to analyze. To consider a motivating example, suppose that each observation consists of a vector of features $X \in \mathbb{R}^p$ and a real-valued response $Y \in \mathbb{R}$. Even in an idealized scenario where we might be certain that the data follow a linear model, it is still not clear how we might perform inference on a bagged prediction obtained by, say, averaging the Lasso predictions on multiple bootstrapped samples of the data.

To address the problem of valid statistical inference for ensemble predictions, we propose a method for constructing a predictive confidence interval for a new observation that can be wrapped around existing ensemble prediction methods. Our method integrates ensemble learning with the recently proposed *jackknife+* [2]. It is implemented by tweaking how the ensemble aggregates the learned predictions. This makes the resulting integrated algorithm to output an interval-valued prediction that, when run at a target predictive coverage level of $1 - \alpha$, provably covers the new response value at least $1 - 2\alpha$ proportion of the time in the worst case, with no assumptions on the data beyond independent and identically distributed samples.

---
[*]Department of Statistics, The University of Chicago, Chicago, IL 60637, USA byolkim@uchicago.edu
[†]H. Milton Stewart School of Industrial & Systems Engineering, Georgia Institute of Technology, Atlanta, GA 30332, USA cxu310@gatech.edu
[‡]Department of Statistics, The University of Chicago, Chicago, IL 60637, USA rina@uchicago.edu

Our main contributions are as follows.

- We propose the jackknife+-after-bootstrap (J+aB), a method for constructing predictive confidence intervals that can be efficiently wrapped around an ensemble learning algorithm chosen by the user.
- We prove that the coverage of a J+aB interval is at worst $1 - 2\alpha$ for the assumption-free theory. This lower bound is non-asymptotic, and holds for any sample size and any distribution of the data.
- We verify that the empirical coverage of a J+aB interval is actually close to $1 - \alpha$.

## 2 Background and related work

Suppose we are given $n$ independent and identically distributed (i.i.d.) observations $(X_1, Y_1), \ldots, (X_n, Y_n) \overset{\text{iid}}{\sim} \mathcal{P}$ from some probability distribution $\mathcal{P}$ on $\mathbb{R}^p \times \mathbb{R}$. Given the available training data, we would like to predict the value of the response $Y_{n+1}$ for a new data point with features $X_{n+1}$, where we assume that $(X_{n+1}, Y_{n+1})$ is drawn from the same probability distribution $\mathcal{P}$. A common framework is to fit a regression model $\widehat{\mu} : \mathbb{R}^p \to \mathbb{R}$ by applying some regression algorithm to the training data $\{(X_i, Y_i)\}_{i=1}^n$, and then predicting $\widehat{\mu}(X_{n+1})$ as our best estimate of the unseen test response $Y_{n+1}$.

However, the question arises: How can we quantify the likely accuracy or error level of these predictions? For example, can we use the available information to build an interval around our estimate $\widehat{\mu}(X_{n+1}) \pm$ (some margin of error) that we believe is likely to contain $Y_{n+1}$? More generally, we want to build a *predictive interval* $\widehat{C}(X_{n+1}) \subseteq \mathbb{R}$ that maps the test features $X_{n+1}$ to an interval (or more generally, a set) believed to contain $Y_{n+1}$. Thus, instead of $\widehat{\mu} : \mathbb{R}^p \to \mathbb{R}$, we would like our method to output $\widehat{C} : \mathbb{R}^p \to \mathbb{R}^2$ with the property

$$\mathbb{P}\left[ Y_{n+1} \in \widehat{C}(X_{n+1}) \right] \geq 1 - \alpha, \tag{1}$$

where the probability is with respect to the distribution of the $n + 1$ training and test data points (as well as any additional source of randomness used in obtaining $\widehat{C}$). Ideally, we want $\widehat{C}$ to satisfy (1) for any data distribution $\mathcal{P}$. Such $\widehat{C}$ is said to satisfy *distribution-free predictive coverage* at level $1 - \alpha$.

### 2.1 Jackknife and jackknife+

One of the methods that can output $\widehat{C}$ with distribution-free predictive coverage is the recent jackknife+ of Barber et al. [2] which inspired our work. As suggested by the name, the jackknife+ is a simple modification of the jackknife approach to constructing predictive confidence intervals.

To define the jackknife and the jackknife+, we begin by introducing some notation. Let $\mathcal{R}$ denote any regression algorithm; $\mathcal{R}$ takes in a training data set, and outputs a model $\widehat{\mu} : \mathbb{R}^p \to \mathbb{R}$, which can then be used to map a new $X$ to a predicted $Y$. We will write $\widehat{\mu} = \mathcal{R}(\{(X_i, Y_i)\}_{i=1}^n)$ for the model fitted on the full training data, and will also write $\widehat{\mu}_{\backslash i} = \mathcal{R}(\{(X_j, Y_j)\}_{j=1, j \neq i}^n)$ for the model fitted on the training data without the point $i$. Let $q_{\alpha,n}^+\{v_i\}$ and $q_{\alpha,n}^-\{v_i\}$ denote the upper and the lower $\alpha$-quantiles of a list of $n$ values indexed by $i$, that is to say, $q_{\alpha,n}^+\{v_i\} =$ the $\lceil (1 - \alpha)(n + 1) \rceil$-th smallest value of $v_1, \ldots, v_n$, and $q_{\alpha,n}^-\{v_i\} = -q_{\alpha,n}^+\{-v_i\}$.

The jackknife prediction interval is given by

$$\widehat{C}_{\alpha,n}^{\text{J}}(x) = \widehat{\mu}(x) \pm q_{\alpha,n}^+\{R_i\} = \left[ q_{\alpha,n}^-\{\widehat{\mu}(x) - R_i\}, q_{\alpha,n}^+\{\widehat{\mu}(x) + R_i\} \right], \tag{2}$$

where $R_i = |Y_i - \widehat{\mu}_{\backslash i}(X_i)|$ is the $i$-th leave-one-out residual. This is based on the idea that the $R_i$'s are good estimates of the test residual $|Y_{n+1} - \widehat{\mu}_{\backslash i}(X_{n+1})|$, because the data used to train $\widehat{\mu}_{\backslash i}$ is independent of $(X_i, Y_i)$. Perhaps surprisingly, it turns out that fully assumption-free theory is impossible for (2) [see 2, Theorem 2]. By contrast, it is achieved by the jackknife+, which modifies (2) by replacing $\widehat{\mu}$ with $\widehat{\mu}_{\backslash i}$'s:

$$\widehat{C}_{\alpha,n}^{\text{J+}}(x) = \left[ q_{\alpha,n}^-\{\widehat{\mu}_{\backslash i}(x) - R_i\}, q_{\alpha,n}^+\{\widehat{\mu}_{\backslash i}(x) + R_i\} \right]. \tag{3}$$

Barber et al. [2] showed that (3) satisfies distribution-free predictive coverage at level $1 - 2\alpha$. Intuitively, the reason that such a guarantee is impossible for (2) is that the test residual $|Y_{n+1} - \widehat{\mu}(X_{n+1})|$ is not quite comparable with the leave-one-out residuals $|Y_i - \widehat{\mu}_{\backslash i}(X_i)|$, because $\widehat{\mu}$ always sees one more observation in training than $\widehat{\mu}_{\backslash i}$ sees. The jackknife+ correction restores the symmetry, making assumption-free theory possible.

## 2.2 Ensemble methods

In this paper, we are concerned with ensemble predictions that apply a base regression method $\mathcal{R}$, such as linear regression or the Lasso, to different training sets generated from the training data by a resampling procedure.

Specifically, the ensemble method starts by creating multiple training data sets (or multisets) of size $m$ from the available training data points $\{1, \ldots, n\}$. We may choose the sets by *bootstrapping* (sampling $m$ indices uniformly at random with replacement—a typical choice is $m = n$), or by *subsampling* (sampling without replacement, for instance with $m = n/2$).

For each $b$, the algorithm calls on $\mathcal{R}$ to fit the model $\widehat{\mu}_b$ using the training set $S_b$, and then aggregates the $B$ predictions $\widehat{\mu}_1(x), \ldots, \widehat{\mu}_B(x)$ into a single final prediction $\widehat{\mu}_\varphi(x)$ via an aggregation function $\varphi$,[4] typically chosen to be a simple function such as the median, mean, or trimmed mean. When $\varphi$ is the mean, the ensemble method run with bootstrapped $S_b$'s is referred to as *bagging* [5], while if we instead use subsampled $S_b$'s, then this ensembling procedure is referred to as *subagging* [8].

The procedure is formalized in Algorithm 1.

---

**Algorithm 1** Ensemble learning

---

**Input:** Data $\{(X_i, Y_i)\}_{i=1}^n$
**Output:** Ensembled regression function $\widehat{\mu}_\varphi$
    **for** $b = 1, \ldots, B$ **do**
        Draw $S_b = (i_{b,1}, \ldots, i_{b,m})$ by sampling with or without replacement from $\{1, \ldots, n\}$.
        Compute $\widehat{\mu}_b = \mathcal{R}((X_{i_{b,1}}, Y_{i_{b,1}}), \ldots, (X_{i_{b,m}}, Y_{i_{b,m}}))$.
    **end for**
    Define $\widehat{\mu}_\varphi = \varphi(\widehat{\mu}_1, \ldots, \widehat{\mu}_B)$.

---

**Can we apply the jackknife+ to an ensemble method?** While ensembling is generally understood to provide a more robust and stable prediction as compared to the underlying base algorithm, there are substantial difficulties in developing inference procedures for ensemble methods with theoretical guarantees. For one thing, ensemble methods are frequently used with highly discontinuous and nonlinear base learners, and aggregating many of them leads to models that defy an easy analysis. The problem is compounded by the fact that ensemble methods are typically employed in settings where good generative models of the data distribution are either unavailable or difficult to obtain. This makes distribution-free methods that can wrap around arbitrary machine learning algorithms, such as the conformal prediction [36, 18], the split conformal [24, 35, 18], or cross-validation or jackknife type methods [2] attractive, as they retain validity over any data distribution. However, when deployed with ensemble prediction methods which often require a significant overhead from the extra cost of model fitting, the resulting combined procedures tend to be extremely computationally intensive, making them impractical in most settings. In the case of the jackknife+, if each ensembled model makes $B$ many calls to the base regression method $\mathcal{R}$, the jackknife+ would require a total of $Bn$ calls to $\mathcal{R}$. By contrast, our method will require only $O(B)$ many calls to $\mathcal{R}$, making the computational burden comparable to obtaining a single ensemble prediction.

## 2.3 Related work

Our paper contributes to the fast-expanding literature on distribution-free predictive inference, which has garnered attention in recent years for providing valid inferential tools that can work with complex

Table 1: Comparison of computational costs for obtaining $n_{\text{test}}$ predictions

|  | #calls to $\mathcal{R}$ | #evaluations | #calls to $\varphi$ |
|---|---|---|---|
| Ensemble | $B$ | $Bn_{\text{test}}$ | $n_{\text{test}}$ |
| J+ with Ensemble | $Bn$ | $Bn(1 + n_{\text{test}})$ | $n(1 + n_{\text{test}})$ |
| J+aB | $B$ | $B(n + n_{\text{test}})$ | $n(1 + n_{\text{test}})$ |

machine learning algorithms such as neural networks. This is because many of the methods proposed are "wrapper" algorithms that can be used in conjunction with an arbitrary learning procedures. This list includes the conformal prediction methodology of Vovk et al. [36], Lei et al. [18], the split conformal methods explored in Papadopoulos [24], Vovk [35], Lei et al. [18], and the jackknife+ of Barber et al. [2]. Meanwhile, methods such as cross-validation or leave-one-out cross-validation (also called the "jackknife") stabilize the results in practice but require some assumptions to analyze theoretically [33, 34, 2].

The method we propose can also be viewed as a wrapper designed specifically for use with ensemble learners. As mentioned in Section 2.2, applying a distribution-free wrapper around an ensemble prediction method typically results in a combined procedure that is computationally burdensome. This has motivated many authors to come up with cost efficient wrappers for use in the ensemble prediction setting. For example, Papadopoulos et al. [26], Papadopoulos and Haralambous [25] use a holdout set to assess the predictive accuracy of an ensembled model. However, when the sample size $n$ is limited, one may achieve more accurate predictions with a cross-validation or jackknife type method, as such a method avoids reducing the sample size in order to obtain a holdout set. Moreover, by using "out-of-bag" estimates [6], it is often possible to reduce the overall cost to the extent that it is on par with obtaining a single ensemble prediction. This is explored in Johansson et al. [16], where they propose a prediction interval of the form $\widehat{\mu}_\varphi(X_{n+1}) \pm q_{\alpha,n}^+(R_i)$, where $\widehat{\mu}_{\varphi\setminus i} = \varphi(\{\widehat{\mu}_b : b = 1, \ldots, B, S_b \not\ni i\})$ and $R_i = |Y_i - \widehat{\mu}_{\varphi\setminus i}(X_i)|$. Zhang et al. [38] provide a theoretical analysis of this type of prediction interval, ensuring that predictive coverage holds asymptotically under additional assumptions. Devetyarov and Nouretdinov [10], Löfström et al. [20], Boström et al. [4, 3], Linusson et al. [19] study variants of this type of method, but fully distribution-free coverage cannot be guaranteed for these methods. By contrast, our method preserves exchangeability, and hence is able to maintain assumption-free and finite-sample validity.

More recently, Kuchibhotla and Ramdas [17] looked at aggregating conformal inference after subsampling or bootstrapping. Their work proposes ensembling multiple runs of an inference procedure, while in contrast our present work seeks to provide inference for ensembled methods.

Stepping away from distribution-free methods, for the popular random forests [15, 7], Meinshausen [22], Athey et al. [1], Lu and Hardin [21] proposed methods for estimating conditional quantiles, which can be used to construct prediction intervals. The guarantees they provide are necessarily approximate or asymptotic, and rely on additional conditions. Tangentially related are the methods for estimating the variance of the random forest estimator of the conditional mean, e.g., Sexton and Laake [32], Wager et al. [37], Mentch and Hooker [23], which apply, in order, the jackknife-after-bootstrap (not jackknife+) [11] or the infinitesimal jackknife [12] or U-statistics theory. Roy and Larocque [31] propose a heuristic for constructing prediction intervals using such variance estimates. For a comprehensive survey of statistical work related to random forests, we refer the reader to the literature review by Athey et al. [1].

## 3   Jackknife+-after-bootstrap (J+aB)

We present our method, the *jackknife+-after-bootstrap* (J+aB). To design a cost efficient wrapper method suited to the ensemble prediction setting, we borrow an old insight from Breiman [6] and make use of the "out-of-bag" estimates. Specifically, it is possible to obtain the $i$-th leave-one-out model $\widehat{\mu}_{\varphi\setminus i}$ without additional calls to the base regression method by reusing the already computed $\widehat{\mu}_1, \ldots, \widehat{\mu}_B$ by aggregating only those $\widehat{\mu}_b$'s whose underlying training data set $S_b$ did not include the $i$-th data point. This is formalized in Algorithm 2.

---
**Algorithm 2** Jackknife+-after-bootstrap (J+aB)
---
**Input:** Data $\{(X_i, Y_i)\}_{i=1}^n$
**Output:** Predictive interval $\widehat{C}_{\alpha,n,B}^{\text{J+aB}}$
  **for** $b = 1, \ldots, B$ **do**
    Draw $S_b = (i_{b,1}, \ldots, i_{b,m})$ by sampling with or without replacement from $\{1, \ldots, n\}$.
    Compute $\widehat{\mu}_b = \mathcal{R}((X_{i_{b,1}}, Y_{i_{b,1}}), \ldots, (X_{i_{b,m}}, Y_{i_{b,m}}))$.
  **end for**
  **for** $i = 1, \ldots, n$ **do**
    Aggregate $\widehat{\mu}_{\varphi \setminus i} = \varphi(\{\widehat{\mu}_b : b = 1, \ldots, B, \ S_b \not\ni i\})$.
    Compute the residual, $R_i = |Y_i - \widehat{\mu}_{\varphi \setminus i}(X_i)|$.
  **end for**
  Compute the J+aB prediction interval: at each $x \in \mathbb{R}$,

$$\widehat{C}_{\alpha,n,B}^{\text{J+aB}}(x) = \left[ q_{\alpha,n}^- \{\widehat{\mu}_{\varphi \setminus i}(x) - R_i\}, q_{\alpha,n}^+ \{\widehat{\mu}_{\varphi \setminus i}(x) + R_i\} \right].$$

---

Because the J+aB algorithm recycles the *same* $B$ models $\widehat{\mu}_1, \ldots, \widehat{\mu}_B$ to compute all $n$ leave-one-out models $\widehat{\mu}_{\varphi \setminus i}$, the cost of model fitting is identical for the J+aB algorithm and the ensemble learning. Table 1 compares the computational costs of an ensemble method, the jackknife+ wrapped around an ensemble, and the J+aB when the goal is to make $n_{\text{test}}$ predictions. In settings where both model evaluations and aggregations remain relatively cheap, our J+aB algorithm is able to output a more informative confidence interval at virtually no extra cost beyond what is already necessary to produce a single ensemble point prediction. Thus, one can view the J+aB as offering predictive inference "free of charge."

## 4 Theory

In this section, we prove that the coverage of a J+aB interval satisfies a distribution-free lower-bound of $1 - 2\alpha$ in the worst-case. We make two assumptions, one on the data distribution and the other on the ensemble algorithm.

**Assumption 1.** $(X_1, Y_1), \ldots, (X_n, Y_n), (X_{n+1}, Y_{n+1}) \stackrel{\text{iid}}{\sim} \mathcal{P}$, where $\mathcal{P}$ is any distribution on $\mathbb{R}^p \times \mathbb{R}$.

**Assumption 2.** For $k \geq 1$, any fixed $k$-tuple $((x_1, y_1), \ldots, (x_k, y_k)) \in \mathbb{R}^p \times \mathbb{R}$, and any permutation $\sigma$ on $\{1, \ldots, k\}$, it holds that $\mathcal{R}((x_1, y_1), \ldots, (x_k, y_k)) = \mathcal{R}((x_{\sigma(1)}, y_{\sigma(1)}), \ldots, (x_{\sigma(k)}, y_{\sigma(k)}))$ and $\varphi(y_1, \ldots, y_k) = \varphi(y_{\sigma(1)}, \ldots, y_{\sigma(k)})$. In other words, the base regression algorithm $\mathcal{R}$ and the aggregation $\varphi$ are both invariant to the ordering of the input arguments.[5]

Assumption 1 is fairly standard in the distribution-free prediction literature [36, 18, 2]. In fact, our results only require exchangeability of the $n + 1$ data points, as is typical in distribution-free inference—the i.i.d. assumption is a familiar special case. Assumption 2 is a natural condition in the setting where the data points are i.i.d., and therefore should logically be treated symmetrically.

Theorem 1 gives the distribution-free coverage guarantee for the J+aB prediction interval with one intriguing twist: the total number of base models, $B$, must be drawn at *random* rather than chosen in advance. This is because Algorithm 2 as given subtly violates symmetry even when $\mathcal{R}$ and $\varphi$ are themselves symmetric. However, we shall see that requiring $B$ to be Binomial is enough to restore symmetry, after which assumption-free theory is possible.

**Theorem 1.** *Fix any integers $\widetilde{B} \geq 1$ and $m \geq 1$, any base algorithm $\mathcal{R}$, and any aggregation function $\varphi$. Suppose the jackknife+-after-bootstrap method (Algorithm 2) is run with (i) $B \sim$ Binomial$(\widetilde{B}, (1 - \frac{1}{n+1})^m)$ in the case of sampling with replacement or (ii) $B \sim$ Binomial$(\widetilde{B}, 1 - \frac{m}{n+1})$ in the case of sampling without replacement. Then, under Assumptions 1 and 2, the jackknife+-*

*after-bootstrap prediction interval satisfies*

$$\mathbb{P}\left[Y_{n+1} \in \widehat{C}_{\alpha,n,B}^{\text{J+aB}}(X_{n+1})\right] \geq 1 - 2\alpha,$$

*where the probability holds with respect to the random draw of the training data* $(X_1, Y_1), \ldots, (X_n, Y_n)$, *the test data point* $(X_{n+1}, Y_{n+1})$, *and the Binomial* $B$.

*Proof sketch.* Our proof follows the main ideas of the jackknife+ guarantee [2, Theorem 1]. It is a consequence of the jackknife+ construction that the guarantee can be obtained by a simultaneous comparison of all $n$ pairs of leave-one-out(-of-$n$) residuals, $|Y_{n+1} - \widehat{\mu}_{\backslash i}(X_{n+1})|$ vs $|Y_i - \widehat{\mu}_{\backslash i}(X_i)|$ for $i = 1, \ldots, n$. The key insight provided by Barber et al. [2] is that this is easily done by regarding the residuals as leave-*two*-out(-of-$(n+1)$) residuals $|Y_i - \widetilde{\mu}_{\backslash i,j}(X_i)|$ with $\{i, j\} \ni (n+1)$, where $\widetilde{\mu}_{\backslash i,j}$ is a model trained on the *augmented* data combining both training and test points and then screening out the $i$-th and the $j$-th points, one of which is the test point. These leave-two-out residuals are naturally embedded in an $(n+1) \times (n+1)$ array of all the leave-two-out residuals, $R = [R_{ij} = |Y_i - \widetilde{\mu}_{\backslash i,j}(X_i)| : i \neq j \in \{1, \ldots, n, n+1\}]$. Since the $n+1$ data points in the augmented data are i.i.d., they are exchangeable, and hence so is the array $R$, i.e., the distribution of $R$ is invariant to relabeling of the indices. A simple counting argument then ensures that the jackknife+ interval fails to cover with probability at most $2\alpha$. This is the essence of the jackknife+ proof.

Turning to our J+aB, it may be tempting to define $\widetilde{\mu}_{\varphi \backslash i,j} = \varphi(\{\widehat{\mu}_b : S_b \not\ni i, j\})$, the aggregation of all $\widehat{\mu}_b$'s whose underlying data set $S_b$ excludes $i$ and $j$, and go through with the jackknife+ proof. However, this construction is no longer useful; the corresponding $R$ in this case is no longer exchangeable. This is most easily seen by noting that there are always exactly $B$ many $S_b$'s that do not include the test observation $n+1$, whereas the number of $S_b$'s that do not contain a particular training observation $i \in \{1, \ldots, n\}$ is a random number usually smaller than $B$. The issue here is that the J+aB algorithm as given fails to be symmetric for all $n+1$ data points.

However, just as the jackknife+ symmetrized the jackknife by replacing $\widehat{\mu}$ with $\widehat{\mu}_{\backslash i}$'s, the J+aB can also be symmetrized by merely requiring it to run with a Binomial $B$. To see why, consider the "lifted" Algorithm 3.

---

**Algorithm 3** Lifted J+aB residuals

---

**Input:** Data $\{(X_i, Y_i)\}_{i=1}^{n+1}$
**Output:** Residuals $(R_{ij} : i \neq j \in \{1, \ldots, n+1\})$
    **for** $b = 1, \ldots, \widetilde{B}$ **do**
        Draw $\widetilde{S}_b = (i_{b,1}, \ldots, i_{b,m})$ by sampling with or without replacement from $\{1, \ldots, n+1\}$.
        Compute $\widetilde{\mu}_b = \mathcal{R}((X_{i_{b,1}}, Y_{i_{b,1}}), \ldots, (X_{i_{b,m}}, Y_{i_{b,m}}))$.
    **end for**
    **for** pairs $i \neq j \in \{1, \ldots, n+1\}$ **do**
        Aggregate $\widetilde{\mu}_{\varphi \backslash i,j} = \varphi(\{\widetilde{\mu}_b : \widetilde{S}_b \not\ni i, j\})$.
        Compute the residual, $R_{ij} = |Y_i - \widetilde{\mu}_{\varphi \backslash i,j}(X_i)|$.
    **end for**

---

Because all $n+1$ data points are treated equally by Algorithm 3, the resulting array of residuals $R = [R_{ij} : i \neq j \in \{1, \ldots, n+1\}]$ is again exchangeable. Now, for each $i = 1, \ldots, n+1$, define $\widetilde{\mathcal{E}}_i$ as the event that $\sum_{j \in \{1, \ldots, n+1\} \backslash \{i\}} \mathbb{I}[R_{ij} > R_{ji}] \geq (1-\alpha)(n+1)$. Because of the exchangeability of the array, the same counting argument mentioned above ensures $\mathbb{P}[\widetilde{\mathcal{E}}_{n+1}] \leq 2\alpha$.

To relate the event $\widetilde{\mathcal{E}}_{n+1}$ to the actual J+aB interval $\widehat{C}_{\alpha,n,B}^{\text{J+aB}}(X_{n+1})$ being constructed, we need to couple Algorithms 2 and 3. Let $B = \sum_{b=1}^{\widetilde{B}} \mathbb{I}[\widetilde{S}_b \not\ni n+1]$, the number of $\widetilde{S}_b$'s containing only the training data in the lifted construction, and let $1 \leq b_1 < \cdots < b_B \leq \widetilde{B}$ be the indices of such $\widetilde{S}_b$'s. Note that $B$ is Binomially distributed, as required by the theorem. For each $k = 1, \ldots, B$, define $S_k = \widetilde{S}_{b_k}$. Then, each $S_k$ is an independent uniform draw from $\{1, \ldots, n\}$, with or without replacement. Therefore, we can equivalently consider running Algorithm 2 with these particular $S_1, \ldots, S_B$. Furthermore, this ensures that $\widetilde{\mu}_{\varphi \backslash n+1, i} = \widehat{\mu}_{\varphi \backslash i}$ for each $i$, that is, the leave-one-out models in Algorithm 2 coincide with the leave-two-out models in Algorithm 3. Thus, we have constructed a coupling of the J+aB with its lifted version.

Finally, define $\mathcal{E}_{n+1}$ as the event that $\sum_{i=1}^{n} \mathbb{1}\left[|Y_{n+1} - \widehat{\mu}_{\varphi\backslash i}(X_{n+1})| > R_i\right] \geq (1-\alpha)(n+1)$, where $R_i = |Y_i - \widehat{\mu}_{\varphi\backslash i}(X_i)|$ as before. By the coupling we have constructed, we can see that the event $\mathcal{E}_{n+1}$ is equivalent to the lifted event $\widetilde{\mathcal{E}}_{n+1}$, and thus, $\mathbb{P}[\mathcal{E}_{n+1}] = \mathbb{P}[\widetilde{\mathcal{E}}_{n+1}] \leq 2\alpha$. It can be verified that in the event that the J+aB interval fails to cover, i.e., if $Y_{n+1} \notin \widehat{C}_{\alpha,n,B}^{\text{J+aB}}(X_{n+1})$, the event $\mathcal{E}_{n+1}$ must occur, which concludes the proof. The full version of this proof is given in Supplement A. $\qquad\square$

In most settings where a large number of models are being aggregated, we would not expect the distinction of random vs fixed $B$ to make a meaningful difference to the final output. In Supplement B, we formalize this intuition and give a stability condition on the aggregating map $\varphi$ under which the J+aB has valid coverage for any choice of $B$.

Finally, we remark that although we have exclusively used the regression residuals $|Y_i - \widehat{\mu}_{\backslash i}(X_i)|$ in our exposition for concreteness, our method can also accommodate alternative measures of conformity, e.g., using quantile regression as in Romano et al. [30] or weighted residuals as in Lei et al. [18] which can better handle heteroscedasticity. More generally, if $\widehat{c}_{\varphi\backslash i}$ is the trained conformity measure aggregated from the $S_b$'s that did not use the $i$-th point, then the corresponding J+aB set is given by

$$\widehat{C}_{\alpha,n,B}^{c\text{-J+aB}}(x) = \left\{ y : \sum_{i=1}^{n} \mathbb{1}\left[\widehat{c}_{\varphi\backslash i}(x,y) > \widehat{c}_{\varphi\backslash i}(X_i, Y_i)\right] < (1-\alpha)(n+1) \right\}.$$

## 5 Experiments

In this section, we demonstrate that the J+aB intervals enjoy coverage near the nominal level of $1-\alpha$ numerically, using three real data sets and different ensemble prediction methods. In addition, we also look at the results for the jackknife+, combined either with the same ensemble method (J+ENSEMBLE) or with the non-ensembled base method (J+NON-ENSEMBLE); the precise definitions are given in Supplement D.1. The code is available online.[6]

We used three real data sets, which were also used in Barber et al. [2], following the same data preprocessing steps as described therein. The Communities and Crime (COMMUNITIES) data set [28] contains information on 1994 communities with $p = 99$ covariates. The response $Y$ is the per capita violent crime rate. The BlogFeedback (BLOG) data set [9] contains information on 52397 blog posts with $p = 280$ covariates. The response is the number of comments left on the blog post in the following 24 hours, which we transformed as $Y = \log(1 + \#\text{comments})$. The Medical Expenditure Panel Survey (MEPS) 2016 data set from the Agency for Healthcare Research and Quality, with details for older versions in [13], contains information on 33005 individuals with $p = 107$ covariates. The response is a score measuring each individual's utilization level of medical services. We transformed this as $Y = \log(1 + \text{utilization score})$.

For the base regression method $\mathcal{R}$, we used either the ridge regression (RIDGE), the random forest (RF), or a neural network (NN). For RIDGE, we set the penalty at $\lambda = 0.001\|X\|^2$, where $\|X\|$ is the spectral norm of the training data matrix. RF was implemented using the `RandomForestRegressor` method from `scikit-learn` with 20 trees grown for each random forest using the mean absolute error criterion and the `bootstrap` option turned off, with default settings otherwise. For NN, we used the `MLPRegressor` method from `scikit-learn` with the L-BFGS solver and the logistic activation function, with default settings otherwise. For the aggregation $\varphi$, we used averaging (MEAN). Results obtained with other aggregation methods are discussed in Supplement D.2.

We fixed $\alpha = 0.1$ for the target coverage of 90%. We used $n = 40$ observations for training, sampling uniformly *without* replacement to create a training-test split for each trial. The results presented here are from 10 independent training-test splits of each data set. The ensemble wrappers J+aB and J+ENSEMBLE used sampling *with* replacement. We varied the size $m$ of each bootstrap replicate as $m/n = 0.2, 0.4, \ldots, 1.0$. For J+ENSEMBLE, we used $B = 20$. For the J+aB, we drew $B \sim \text{Binomial}(\widetilde{B}, (1 - \frac{1}{n+1})^m)$ with $\widetilde{B} = [20/\{(1 - \frac{1}{n+1})^m(1 - \frac{1}{n})^m\}]$, where $[\cdot]$ refers to the integer part of the argument. This ensures that the number of models being aggregated for each leave-one-out model is matched on average to the number in J+ENSEMBLE. We remark that the scale

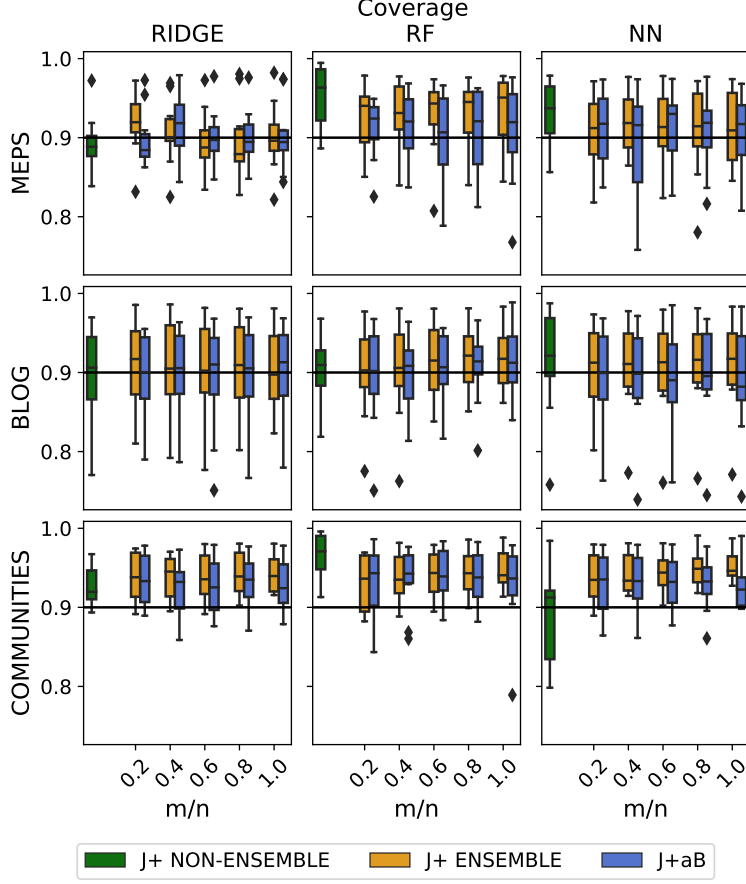

Figure 1: Distributions of coverage (averaged over each test data) in 10 independent splits for $\varphi =$ MEAN. The black line indicates the target coverage of $1 - \alpha$.

of our experiments, as reflected in the number of different training-test splits or the size of $n$ or $B$, has been limited by the computationally inefficient J+ENSEMBLE.

We emphasize that we made no attempt to optimize any of our models. This is because our goal here is to illustrate certain properties of our method that hold *universally* for any data distribution and any ensemble method, and not just in cases when the method happens to be the "right" one for the data. All other things being equal, the statistical efficiency of the intervals our method constructs would be most impacted by how accurately the model is able to capture the data. However, because the method we propose leaves this choice up to the users, performance comparisons along the axis of different ensemble methods are arguably not very meaningful.

We are rather more interested in comparisons of the J+aB and J+ENSEMBLE, and of the J+aB (or J+ENSEMBLE) and J+NON-ENSEMBLE. For the J+aB vs J+ENSEMBLE comparison, we are on the lookout for potential systematic tradeoffs between computational and statistical efficiency. For each $i$, conditional on the event that the same number of models were aggregated for the $i$-th leave-one-out models $\widehat{\mu}_{\varphi \backslash i}$ in the J+aB and J+ENSEMBLE, the two $\widehat{\mu}_{\varphi \backslash i}$'s have the same marginal distribution. However, this is not the case for the joint distribution of all $n$ leave-one-out models $\{\widehat{\mu}_{\varphi \backslash i}\}_{i=1}^{n}$; with respect to the resampling measure, the collection is highly correlated in the case of the J+aB, and independent in the case of J+ENSEMBLE. Thus, in principle, the statistical properties of $\widehat{C}_{\alpha,n,B}^{\text{J+aB}}$ and $\widehat{C}_{\alpha,n,B'}^{\text{J+ENSEMBLE}}$ could differ, although it would be a surprise if it were to turn out that one method always performed better than the other. In comparing the J+aB (or J+ENSEMBLE) and J+NON-ENSEMBLE, we seek to reaffirm some known results in bagging. It is well-known that bagging improves the accuracy of unstable predictors, but has little effect on stable ones [5, 8]. It is reasonable to expect that this property will manifest in some way when the width of $\widehat{C}_{\alpha,n,B}^{\text{J+aB}}$ (or $\widehat{C}_{\alpha,n,B'}^{\text{J+ENSEMBLE}}$) is compared to

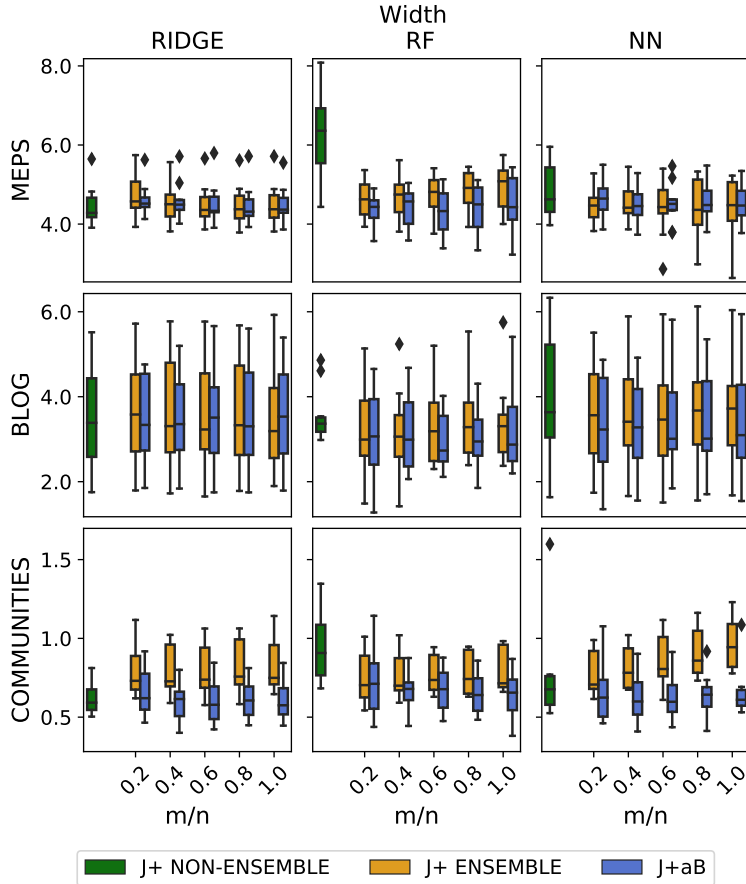

Figure 2: Distributions of interval width (averaged over each test data) in 10 independent splits for $\varphi = \text{MEAN}$.

that of $\widehat{C}_{\alpha,n}^{\text{J+NON-ENSEMBLE}}$. We expect the former to be narrower than the latter when the base regression method is unstable (e.g., RF), but not so when it is already stable (e.g., RIDGE).

Figures 1 and 2 summarize the results of our experiments. First, from Figure 1, it is clear that the coverage of the J+aB is near the nominal level. This is also the case for J+ENSEMBLE or J+NON-ENSEMBLE. Second, in Figure 2, we observe no evidence of a consistent trend of one method always outperforming the other in terms of the precision of the intervals, although we do see some slight variations across different data sets and regression algorithms. Thus, we prefer the computationally efficient J+aB to the costly J+ENSEMBLE. Finally, comparing the J+aB (or J+ENSEMBLE) and J+NON-ENSEMBLE, we find the effect of bagging reflected in the interval widths, and we see improved precision in the case of RF, and for some data sets and at some values of $m$, in the case of NN. Thus, in settings where the base learner is expected to benefit from ensembling, the J+aB is a practical method for obtaining informative prediction intervals that requires a level of computational resources on par with the ensemble algorithm itself.

## 6  Conclusion

We propose the jackknife+-after-bootstrap (J+aB), a computationally efficient wrapper method tailored to the setting of ensemble learning, where by a simple modification to the aggregation stage, the method outputs a predictive interval with fully assumption-free coverage guarantee in place of a point prediction. The J+aB provides a mechanism for quantifying uncertainty in ensemble predictions that is both straightforward to implement and easy to interpret, and can therefore be easily integrated into existing ensemble models.

# 7 Broader impact

Machine learning algorithms are becoming increasingly pervasive in many application areas involving complicated and high-stakes decision making including medical treatment planning and diagnosis, public health, and public policy. As the use of machine learning becomes more widespread, however, we are also becoming more cognizant of the potential pitfalls due to hidden biases in the data or unexpected behavior of blackbox algorithms. Quantifying the uncertainty in machine predictions is one way to safeguard against such errors. Doing so in a meaningful way without making unverifiable or overly simplistic assumptions is a challenge, as data in these applications often exhibit complex phenomena, such as censoring or missingness, heavy tails, multi-modality, etc.. Methods such as our J+aB provide predictive inference guarantees that can be efficiently implemented on large scale data sets under very few assumptions.

## Acknowledgments and Disclosure of Funding

R.F.B. was supported by the National Science Foundation via grant DMS-1654076. The authors are grateful to Aaditya Ramdas and Chirag Gupta for helpful suggestions on related works.

## Footnotes

[4]Formally, we define $\varphi$ as a map from $\bigcup_{k \geq 0} \mathbb{R}^k \to \mathbb{R}$, mapping any collection of predictions in $\mathbb{R}$ to a single aggregated prediction. (If the collection is empty, we would simply output zero or some other default choice). $\varphi$ lifts naturally to a map on vectors of functions, by writing $\widehat{\mu}_\varphi = \varphi(\widehat{\mu}_1, \ldots, \widehat{\mu}_B)$, where $\widehat{\mu}_\varphi(x)$ is defined for each $x \in \mathbb{R}$ by applying $\varphi$ to the collection $(\widehat{\mu}_1(x), \ldots, \widehat{\mu}_B(x))$.

[5]If $\mathcal{R}$ and/or $\varphi$ involve any randomization—for example if $\varphi$ operates by sampling from the collection of predictions—then we can require that the outputs are equal in distribution under any permutation of the input arguments, rather than requiring that equality holds deterministically. In this case, the coverage guarantees in our theorems hold on average over the randomization in $\mathcal{R}$ and/or $\varphi$, in addition to the distribution of the data.

[6]https://www.stat.uchicago.edu/~rina/jackknife+-after-bootstrap_realdata.html

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
