[Supplementary Material]

# Supplementary Material for: Predictive Inference Is Free with the Jackknife+-after-Bootstrap

## A  Proof of Theorem 1

For completeness, we give the full details of the proof of Theorem 1; a sketch of the proof is presented in Section 4 of the main paper.

Denote Algorithm 3 by $\tilde{\mathcal{A}}$. We view $\tilde{\mathcal{A}}$ as mapping a given input $\{(X_i, Y_i)\}_{i=1}^{n+1}$ and a collection of subsamples or bootstrapped samples $\tilde{S}_1, \ldots, \tilde{S}_B$ to a matrix of residuals $R \in \mathbb{R}^{(n+1)\times(n+1)}$, where

$$R_{ij} = \begin{cases} \left| Y_i - \tilde{\mu}_{\varphi\backslash i,j}(X_i) \right| & \text{if } i \neq j, \\ 0 & \text{if } i = j. \end{cases}$$

For any permutation $\sigma$ on $\{1, \ldots, n+1\}$, let $\Pi_\sigma$ stand for its matrix representation—that is, $\Pi_\sigma \in \{0,1\}^{(n+1)\times(n+1)}$ has entries $(\Pi_\sigma)_{\sigma(i),i} = 1$ for each $i$, and zeros elsewhere. Furthermore, for each subsample or bootstrapped sample $\tilde{S}_b = \{i_{b,1}, \ldots, i_{b,m}\}$, write $\sigma(\tilde{S}_b) = \{\sigma(i_{b,1}), \ldots, \sigma(i_{b,m})\}$.

We now claim that

$$R \overset{d}{=} \Pi_\sigma R \Pi_\sigma^\top, \tag{S1}$$

for any fixed permutation $\sigma$ on $\{1, \ldots, n+1\}$. Here $R$ is the residual matrix obtained by a run of Algorithm 3, namely,

$$R = \tilde{\mathcal{A}}\left( (X_1, Y_1), \ldots, (X_{n+1}, Y_{n+1}); \tilde{S}_1, \ldots, \tilde{S}_B \right).$$

To see why (S1) holds, observe that deterministically, we have

$$\Pi_\sigma R \Pi_\sigma^\top = \tilde{\mathcal{A}}\left( (X_{\sigma(1)}, Y_{\sigma(1)}), \ldots, (X_{\sigma(n+1)}, Y_{\sigma(n+1)}); \sigma(\tilde{S}_1), \ldots, \sigma(\tilde{S}_B) \right).$$

Furthermore, we have

$$\left( (X_1, Y_1), \ldots, (X_{n+1}, Y_{n+1}) \right) \overset{d}{=} \left( (X_{\sigma(1)}, Y_{\sigma(1)}), \ldots, (X_{\sigma(n+1)}, Y_{\sigma(n+1)}) \right)$$

by Assumption 1, and

$$\left( \tilde{S}_1, \ldots, \tilde{S}_B \right) \overset{d}{=} \left( \sigma(\tilde{S}_1), \ldots, \sigma(\tilde{S}_B) \right)$$

since subsampling or resampling treats all the indices the same. Finally, the subsamples or bootstrapped samples (i.e., the $\tilde{S}_b$'s) are drawn independently of the data points (i.e., the $(X_i, Y_i)$'s). Combining these calculations yields (S1).

Next, given $R$, define a "tournament matrix" $A = A(R)$ as

$$A_{ij} = \begin{cases} \mathbb{1}\left[R_{ij} > R_{ji}\right] & \text{if } i \neq j, \\ 0 & \text{if } i = j. \end{cases}$$

19 It is easily checked that $A(\Pi_\sigma R \Pi_\sigma^\top) = \Pi_\sigma A(R) \Pi_\sigma^\top$, and hence (S1) implies that

$$A \overset{d}{=} \Pi_\sigma A \Pi_\sigma^\top. \tag{S2}$$

20 Let $S_\alpha(A)$ be the set of row indices with row sums greater than or equal to $(1-\alpha)(n+1)$, i.e.,

$$S_\alpha(A) = \left\{ i = 1, \ldots, n+1 : \sum_{j=1}^{n+1} A_{ij} \geq (1-\alpha)(n+1) \right\}.$$

21 The argument of Step 3 in the proof of Barber et al. [1, Theorem 1] applies to the lifted J+aB
22 "tournament matrix" $A$, and it holds deterministically that

$$|S_\alpha(A)| \leq 2\alpha(n+1). \tag{S3}$$

23 On the other hand, if $j$ is any index, and $\sigma$ is any permutation that swaps indices $n+1$ and $j$, then

$$\mathbb{P}\big[n+1 \in S_\alpha(A)\big] = \mathbb{P}\big[j \in S_\alpha(\Pi_\sigma A \Pi_\sigma^\top)\big] = \mathbb{P}\big[j \in S_\alpha(A)\big].$$

24 The first two events are the same, and the second equality uses (S2). Thus,

$$\mathbb{P}\big[n+1 \in S_\alpha(A)\big] = \frac{1}{n+1} \sum_{j=1}^{n+1} \mathbb{P}\big[j \in S_\alpha(A)\big]$$

$$= \frac{1}{n+1} \mathbb{E}\left[ \sum_{j=1}^{n+1} \mathbb{1}\big[j \in S_\alpha(A)\big] \right] = \frac{\mathbb{E}|S_\alpha(A)|}{n+1} \leq 2\alpha. \tag{S4}$$

25 Note that the event $\big[n+1 \in S_\alpha(A)\big]$ is exactly the event $\tilde{\mathcal{E}}_{n+1}$, defined in Section 4. As described in
26 the proof sketch in Section 4 of the main paper, we can couple this lifted event to the event $\mathcal{E}_{n+1}$,
27 also defined in Section 4 in terms of the actual J+aB, as follows. Let $B = \sum_{b=1}^{\tilde{B}} \mathbb{1}\big[\tilde{S}_b \not\ni n+1\big]$, the
28 number of $\tilde{S}_b$'s containing only training data, and let $1 \leq b_1 < \cdots < b_B \leq \tilde{B}$ be the corresponding
29 indices. Note that the distribution of $B$ is Binomial, as specified in the theorem. Now, for each
30 $k = 1, \ldots, B$, define $S_k = \tilde{S}_{b_k}$. We can observe that each $S_k$ is an independent uniform draw
31 from $\{1, \ldots, n\}$ (with or without replacement). Therefore, we can equivalently consider running
32 J+aB (Algorithm 2) with these particular subsamples or bootstrapped samples $S_1, \ldots, S_B$, in which
33 case it holds deterministically that $\tilde{\mu}_{\varphi \backslash n+1, i} = \hat{\mu}_{\varphi \backslash i}$ for each $i = 1, \ldots, n$. This ensures that
34 $|Y_{n+1} - \tilde{\mu}_{\varphi \backslash n+1, i}(X_{n+1})| = |Y_{n+1} - \hat{\mu}_{\varphi \backslash i}(X_{n+1})|$ and $|Y_i - \tilde{\mu}_{\varphi \backslash i, n+1}(X_i)| = |Y_i - \hat{\mu}_{\varphi \backslash i}(X_i)|$,
35 and thus,
$$\mathbb{P}[\mathcal{E}_{n+1}] = \mathbb{P}[\tilde{\mathcal{E}}_{n+1}] \leq 2\alpha.$$

36 Finally, as in Step 1 in the proof of Barber et al. [1, Theorem 1], it easily follows from the definition
37 of $\hat{C}_{\alpha,n,B}^{\text{J+aB}}$ that if $Y_{n+1} \notin \hat{C}_{\alpha,n,B}^{\text{J+aB}}(X_{n+1})$ then the event $\mathcal{E}_{n+1}$ must occur. Indeed, if $Y_{n+1} \notin$
38 $\hat{C}_{\alpha,n,B}^{\text{J+aB}}(X_{n+1})$, then either $Y_{n+1}$ falls below the lower bound, i.e.,

$$\sum_{i=1}^{n} \mathbb{1}\left[ Y_{n+1} - \hat{\mu}_{\varphi \backslash i}(X_{n+1}) < \big|Y_i - \hat{\mu}_{\varphi \backslash i}(X_i)\big| \right] \geq (1-\alpha)(n+1),$$

39 or $Y_{n+1}$ exceeds the upper bound, i.e.,

$$\sum_{i=1}^{n} \mathbb{1}\left[ Y_{n+1} - \hat{\mu}_{\varphi \backslash i}(X_{n+1}) > \big|Y_i - \hat{\mu}_{\varphi \backslash i}(X_i)\big| \right] \geq (1-\alpha)(n+1),$$

40 and the above two expressions imply

$$\sum_{i=1}^{n} \mathbb{1}\left[ \big|Y_{n+1} - \hat{\mu}_{\varphi \backslash i}(X_{n+1})\big| > \big|Y_i - \hat{\mu}_{\varphi \backslash i}(X_i)\big| \right] \geq (1-\alpha)(n+1).$$

41 Therefore, we conclude that

$$\mathbb{P}\left[ Y_{n+1} \notin \hat{C}_{\alpha,n,B}^{\text{J+aB}}(X_{n+1}) \right] \leq 2\alpha,$$

42 thus proving the theorem.

## B  Guarantees with stability

Many ensembles that are used in practice are variants of bagging, where multiple independent copies of the given training data set are generated through a resampling mechanism, after which estimates from different data sets are pooled together via an averaging procedure of some kind. Bagging can be understood as a smoothing operation that when applied on a discontinuous base learner, often greatly improve its accuracy [3–5].

For ensembles of this type, the aggregated predictions they produce frequently exhibit a concentrating behavior as $B \to \infty$, making the corresponding J+aB interval much like a jackknife+ interval. In such cases, it is reasonable to expect a J+aB interval to remain valid regardless of the choice of $B$, e.g., random with a Binomial distribution or fixed, by its proximity to a jackknife+ interval. Intuitively, this happens when the aggregation is insensitive to any one prediction participating in the ensemble.

To formalize, let $\mathbb{E}^*$ denote the expectation with respect to the resampling measure — that is, we take the expectation with respect to the random collection of subsamples or bootstrapped samples $S_1, \ldots, S_B$ conditional on all the observed data $\{(X_i, Y_i)\}_{i=1}^n$ and $X_{n+1}$. For example, when $\varphi(\cdot) = \mathrm{mean}(\cdot)$ is the mean aggregation,

$$\mathbb{E}^* [\hat{\mu}_{\mathrm{mean}}(X_{n+1})] = \mathbb{E}\left[\hat{\mu}_1(X_{n+1}) \big| (X_1, Y_1), \ldots, (X_n, Y_n), X_{n+1}\right],$$

the expected prediction from the model $\hat{\mu}_1$ fitted on training sample $S_1$, where the expectation is taken with respect to the draw of $S_1$.

**Assumption S1** (Ensemble stability)**.** For $\varepsilon \geq 0$ and $\delta \in (0, 1)$, it holds for each $i = 1, \ldots, n$ that

$$\mathbb{P}\left[\left|\hat{\mu}_{\varphi \backslash i}(X_i) - \mathbb{E}^*\left[\hat{\mu}_{\varphi \backslash i}(X_i)\right]\right| > \varepsilon\right] \leq \delta.$$

Here $\hat{\mu}_{\varphi \backslash i}$ is the ensembled leave-one-out model defined in Algorithm 2. To gain intuition for this assumption, we consider the mean aggregation as a canonical example, and verify that it satisfies Assumption S1 for any bounded base regression method.

**Proposition S1.** *Suppose that $\varphi(\cdot) = \mathrm{mean}(\cdot)$ is the mean aggregation, and suppose the base regression method $\mathcal{R}$ always outputs a bounded regression function, i.e., $\mathcal{R}$ maps any training data set to a function $\hat{\mu}$ taking values in a bounded range $[\ell, u]$, for fixed constants $\ell < u$. Then, for any $\varepsilon > 0$, Assumption S1 is satisfied with*

$$\delta = 2 \exp\left(-\frac{2\sqrt{B}\theta\varepsilon^2}{(u - \ell)^2}\right) + \exp\left(-\frac{\left(\sqrt{B} - 1\right)^2 \theta^2}{2}\right),$$

*where $\theta = (1 - \frac{1}{n})^m$ in the case of bagging (i.e., the $S_b$'s are bootstrapped samples, drawn with replacement), or $\theta = 1 - \frac{m}{n}$ in the case of subagging (i.e., the $S_b$'s are subsamples drawn without replacement).*

*Proof.* By exchangeability, it suffices to prove the statement for a single $i \in \{1, \ldots, n\}$. Fix $i$, and let $B_i$ denote the number of $S_b$'s *not* containing the index $i$, i.e., $B_i = \sum_{b=1}^B \mathbb{1}\left[S_b \not\ni i\right]$. For any fixed $\gamma \in (0, 1)$,

$$\mathbb{P}^*\left[\left|\hat{\mu}_{\mathrm{mean}\backslash i}(X_i) - \mathbb{E}^*[\hat{\mu}_{\mathrm{mean}\backslash i}(X_i)]\right| > \varepsilon\right]$$
$$\leq \mathbb{P}^*\left[\left|\hat{\mu}_{\mathrm{mean}\backslash i}(X_i) - \mathbb{E}^*[\hat{\mu}_{\mathrm{mean}\backslash i}(X_i)]\right| > \varepsilon \text{ and } B_i \geq \gamma\theta B\right] + \mathbb{P}^*\left[B_i < \gamma\theta B\right].$$

As for our earlier notation $\mathbb{E}^*$, here $\mathbb{P}^*$ denotes the probability with respect to the random collection of subsamples or bootstrapped samples $S_1, \ldots, S_B$ conditional on the data $(X_1, Y_1), \ldots, (X_n, Y_n)$.

The arithmetic mean aggregation function, $\varphi_{\mathrm{mean}}$, satisfies

$$\sup_{\substack{y_1, \ldots, y_{B_i}, \\ y_b' \in [\ell, u]}} |\varphi_{\mathrm{mean}}(y_1, \ldots, y_{B_i}) - \varphi_{\mathrm{mean}}(y_1, \ldots, y_{b-1}, y_b', y_{b+1}, \ldots, y_{B_i})| \leq \frac{u - \ell}{B_i}, \quad b = 1, \ldots, B_i.$$

Thus, by McDiarmid's inequality [2, Theorem 6.2],

$$\mathbb{P}^*\left[\left|\hat{\mu}_{\mathrm{mean}\backslash i}(X_i) - \mathbb{E}^*[\hat{\mu}_{\mathrm{mean}\backslash i}(X_i)]\right| > \varepsilon \ \middle| \ B_i \geq \gamma\theta B\right] \leq 2 \exp\left(-\frac{2B\gamma\theta\varepsilon^2}{(u - \ell)^2}\right). \quad \text{(S5)}$$

On the other hand, $B_i \sim \text{Binomial}(B, \theta)$, where $\theta = \left(1 - \frac{1}{n}\right)^m$ for sampling with replacement, or $\theta = 1 - \frac{m}{n}$ for sampling without replacement. The Chernoff inequality for the binomial [2, Chapter 2] implies

$$\mathbb{P}\left[B_i < \gamma\theta B\right] \leq \exp\left(-\frac{B\left(1 - \gamma\right)^2 \theta^2}{2}\right). \tag{S6}$$

Combining (S5) and (S6),

$$\mathbb{P}^*\left[\left|\hat{\mu}_{\text{mean}\setminus i}(X_i) - \mathbb{E}^*[\hat{\mu}_{\text{mean}\setminus i}(X_i)]\right| > \varepsilon\right] \leq 2\exp\left(-\frac{2B\gamma\theta\varepsilon^2}{(u - \ell)^2}\right) + \exp\left(-\frac{B\left(1 - \gamma\right)^2 \theta^2}{2}\right).$$

Taking $\gamma = 1/\sqrt{B}$ yields

$$\mathbb{P}^*\left[\left|\hat{\mu}_{\text{mean}\setminus i}(X_i) - \mathbb{E}^*[\hat{\mu}_{\text{mean}\setminus i}(X_i)]\right| > \varepsilon\right] \leq 2\exp\left(-\frac{2\sqrt{B}\theta\varepsilon^2}{(u - \ell)^2}\right) + \exp\left(-\frac{\left(\sqrt{B} - 1\right)^2 \theta^2}{2}\right).$$

$\square$

To study coverage properties under this notion of stability, we first define the $\varepsilon$-inflated J+aB prediction interval as

$$\hat{C}_{\alpha,n,B}^{\varepsilon\text{-J+aB}}(x) = \left[q_{\alpha,n}^-\{\hat{\mu}_{\varphi\setminus i}(x) - R_i\} - \varepsilon, q_{\alpha,n}^+\{\hat{\mu}_{\varphi\setminus i}(x) + R_i\} + \varepsilon\right],$$

for any $\varepsilon \geq 0$. We then have the following guarantee:

**Theorem S1.** *Under $(\varepsilon, \delta)$-ensemble stability (Assumption S1), the $2\varepsilon$-inflated jackknife+-after-bootstrap prediction interval satisfies*

$$\mathbb{P}\left[Y_{n+1} \in \hat{C}_{\alpha,n,B}^{2\varepsilon\text{-J+aB}}(X_{n+1})\right] \geq 1 - 2\alpha - 4\sqrt{\delta}.$$

Delaying the proof to the end of this section, we discuss the difference between Theorem S1 and Theorem 1. Theorem 1 gives an *assumption-free* lower-bound of $1 - 2\alpha$ on the coverage, but the probability is over all randomness, including that of the Binomial draw. By contrast, the $\approx 1 - 2\alpha$ coverage guarantee of Theorem S1 holds for a *fixed* value of $B$ used to run Algorithm 2, but at the cost of requiring the ensemble algorithm $\mathcal{R}_{\varphi,B}$ to satisfy ensemble stability.

In contrast to the above notion of ensemble stability, Steinberger and Leeb [6] and Barber et al. [1] study coverage of jackknife and jackknife+ under *algorithmic stability* of (non-ensembled) regression method $\mathcal{R}$. This requires $\mathcal{R}$ to satisfy

$$\mathbb{P}\left[\left|\hat{\mu}_{\setminus i}(X_{n+1}) - \hat{\mu}(X_{n+1})\right| > \varepsilon^*\right] \leq \delta^*. \tag{S7}$$

This can be interpreted as saying that a prediction $\hat{\mu}(X_{n+1})$ is only slightly perturbed if a single point is removed from the training. In this setting, jackknife and jackknife+ are each shown to guarantee $\approx 1 - \alpha$ coverage.

We can take a lifted version of this assumption, requiring that (S7) holds on the ensembled models on average over the resampling process:

$$\mathbb{P}\left[\left|\mathbb{E}^*\left[\hat{\mu}_{\varphi\setminus i}(X_{n+1}) - \mathbb{E}^*\left[\hat{\mu}_{\varphi}(X_{n+1})\right]\right]\right| > \varepsilon^*\right] \leq \delta^*. \tag{S8}$$

Note that one can have ensemble stability without algorithmic stability. For example, a bounded regression method may still be highly unstable relative to adding/removing a single data point (thus violating algorithmic stability), while Proposition S1 ensures that ensemble stability will hold under mean aggregation.

When an ensemble method satisfies both Assumption S1 and the lifted version of algorithmic stability (S8), then the following result yields a coverage bound that is $\approx 1 - \alpha$, rather than $\approx 1 - 2\alpha$ as in Theorem S1:

**Theorem S2.** *Assume that $(\varepsilon, \delta)$-ensemble stability (Assumption S1) holds, and in addition, the ensembled model satisfies algorithmic stability on average over the resampling process, i.e., (S8). Then the $2\varepsilon + 2\varepsilon^*$-inflated J+aB prediction interval satisfies*

$$\mathbb{P}\left[Y_{n+1} \in \hat{C}_{\alpha,n,B}^{(2\varepsilon+2\varepsilon^*)\text{-J+aB}}(X_{n+1})\right] \geq 1 - \alpha - 3\sqrt{\delta} - 4\sqrt{\delta^*}.$$

112 *Proof of Theorems S1 and S2.* Put $\hat{\mu}^*_{\varphi \setminus i} = \mathbb{E}^*[\hat{\mu}_{\varphi \setminus i}]$, where we recall that $\mathbb{E}^*$ is the expectation
113 conditional on the data. Let $\mathcal{R}^*_\varphi$ denote the regression algorithm mapping data to $\hat{\mu}^*_\varphi$, i.e.,

$$\mathcal{R}^*_\varphi : \{(X_i, Y_i)\}^n_{i=1} \mapsto \mathbb{E}^* \left[ \varphi \left( \{ \mathcal{R} \left( \{(X_{i_{b,\ell}}, Y_{i_{b,\ell}})\}^m_{\ell=1} \right) : b = 1, \ldots, B', B' \sim \mathrm{Binomial}(B, \theta) \} \right) \right],$$

114 where $\theta = \theta(n) = (1 - \frac{1}{n+1})^m$ (in the case of sampling with replacement) or $\theta = \theta(n) = 1 - \frac{m}{n+1}$
115 (in the case of sampling without replacement). We emphasize that $n$ here refers to the size of the
116 sample being fed through $\mathcal{R}^*_\varphi$ (e.g., each leave-one-out regressor $\hat{\mu}^*_{\varphi \setminus i}$ is trained on $n - 1$ data points,
117 so in this case, $\theta = \theta(n-1)$). $\mathcal{R}^*_\varphi$ is a deterministic function of the data, since it averages over the
118 random draw of the subsamples or bootstrapped samples. Furthermore, it is a symmetric regression
119 algorithm (i.e., satisfies Assumption 2).

120 Fix some $\alpha' \in (0,1)$ to be determined later, and construct the jackknife+ interval

$$\hat{C}^{*\mathrm{J}+}_{\alpha',n}(x) = \left[ q^-_{\alpha',n} \{ \hat{\mu}^*_{\varphi \setminus i}(x) - R^*_i \}, q^+_{\alpha',n} \{ \hat{\mu}^*_{\varphi \setminus i}(x) + R^*_i \} \right],$$

121 where $R^*_i = |Y_i - \hat{\mu}^*_{\varphi \setminus i}(X_i)|$ is the leave-one-out residual for this new regression algorithm. By
122 Barber et al. [1, Theorem 1], $\hat{C}^{*\mathrm{J}+}_{\alpha',n}$ satisfies

$$\mathbb{P} \left[ Y_{n+1} \in \hat{C}^{*\mathrm{J}+}_{\alpha',n}(X_{n+1}) \right] \geq 1 - 2\alpha'.$$

123 If, additionally, $\mathcal{R}^*_\varphi$ satisfies the algorithmic stability condition (S7) given in Section B of the main
124 paper, then by Barber et al. [1, Theorem 5], the $2\varepsilon^*$-inflated jackknife+ interval

$$\hat{C}^{*2\varepsilon^* \mathrm{-J}+}_{\alpha',n}(x) = \left[ q^-_{\alpha',n} \{ \hat{\mu}^*_{\varphi \setminus i}(x) - R^*_i \} - 2\varepsilon^*, q^+_{\alpha',n} \{ \hat{\mu}^*_{\varphi \setminus i}(x) + R^*_i \} + 2\varepsilon^* \right]$$

125 satisfies

$$\mathbb{P} \left[ Y_{n+1} \in \hat{C}^{*2\varepsilon^* \mathrm{-J}+}_{\alpha',n}(X_{n+1}) \right] \geq 1 - \alpha' - 4\sqrt{\delta^*}.$$

126 Next, by Assumption S1, for each $i = 1, \ldots, n$,

$$\mathbb{P} \left[ \left| \hat{\mu}_{\varphi \setminus i}(X_i) - \hat{\mu}^*_{\varphi \setminus i}(X_i) \right| > \varepsilon \right] \leq \delta. \tag{S9}$$

127 Let $\alpha' = \alpha - \sqrt{\delta}$. By the above argument, to prove the theorems, it suffices to show

$$\hat{C}^{2\varepsilon \mathrm{-J}+\mathrm{aB}}_{\alpha,n,B}(X_{n+1}) \supseteq \hat{C}^{*\mathrm{J}+}_{\alpha',n}(X_{n+1}) \quad \text{with probability at least} \quad 1 - 2\sqrt{\delta}$$

128 in order to complete the proof of Theorem S1, or

$$\hat{C}^{(2\varepsilon+2\varepsilon^*) \mathrm{-J}+\mathrm{aB}}_{\alpha,n,B}(X_{n+1}) \supseteq \hat{C}^{*2\varepsilon^* \mathrm{-J}+}_{\alpha',n}(X_{n+1}) \quad \text{with probability at least} \quad 1 - 2\sqrt{\delta}$$

129 in order to complete the proof of Theorem S2. In fact, these two claims are proved identically—we
130 simply need to show that

$$\hat{C}^{(2\varepsilon+2\varepsilon') \mathrm{-J}+\mathrm{aB}}_{\alpha,n,B}(X_{n+1}) \supseteq \hat{C}^{*2\varepsilon' \mathrm{-J}+}_{\alpha',n}(X_{n+1}) \quad \text{with probability at least} \quad 1 - 2\sqrt{\delta} \tag{S10}$$

131 with the choice $\varepsilon' = 0$ for Theorem S1, or $\varepsilon' = \varepsilon^*$ for Theorem S2.

132 To complete the proof, then, we establish the bound (S10). Suppose $\hat{C}^{(2\varepsilon+2\varepsilon') \mathrm{-J}+\mathrm{aB}}_{\alpha,n,B}(X_{n+1}) \not\supseteq$
133 $\hat{C}^{*2\varepsilon' \mathrm{-J}+}_{\alpha',n}(X_{n+1})$. We have that either

$$q^+_{\alpha,n} \left\{ \hat{\mu}_{\varphi \setminus i}(X_{n+1}) + R_i \right\} + 2\varepsilon < q^+_{\alpha',n} \left\{ \hat{\mu}^*_{\varphi \setminus i}(X_{n+1}) + R^*_i \right\}$$

134 or

$$q^-_{\alpha,n} \left\{ \hat{\mu}_{\varphi \setminus i}(X_{n+1}) - R_i \right\} - 2\varepsilon > q^-_{\alpha',n} \left\{ \hat{\mu}^*_{\varphi \setminus i}(X_{n+1}) - R^*_i \right\},$$

135 where $R_i = |Y_i - \hat{\mu}_{\varphi \setminus i}(X_i)|$. As in the proof of Barber et al. [1, Theorem 5], this implies that

$$\left| \hat{\mu}_{\varphi \setminus i}(X_{n+1}) - \hat{\mu}^*_{\varphi \setminus i}(X_{n+1}) \right| + \left| \hat{\mu}_{\varphi \setminus i}(X_i) - \hat{\mu}^*_{\varphi \setminus i}(X_i) \right| > 2\varepsilon$$

for at least $\lceil(1-\alpha)(n+1)\rceil - (\lceil(1-\alpha')(n+1)\rceil - 1) \geq \sqrt{\delta}(n+1)$ many indices $i = 1, \dots, n$. Thus,

$$
\mathbb{P}\left[\hat{C}^{(2\varepsilon+2\varepsilon')\text{-J+aB}}_{\alpha,n,B}(X_{n+1}) \not\supseteq \hat{C}^{*2\varepsilon'\text{-J+}}_{\alpha',n}(X_{n+1})\right]
$$

$$
\leq \mathbb{P}\left[\sum_{i=1}^{n} \mathbb{1}\left[\left|\hat{\mu}_{\varphi\backslash i}(X_{n+1}) - \hat{\mu}^*_{\varphi\backslash i}(X_{n+1})\right| + \left|\hat{\mu}_{\varphi\backslash i}(X_i) - \hat{\mu}^*_{\varphi\backslash i}(X_i)\right| > 2\varepsilon\right] \geq \sqrt{\delta}(n+1)\right]
$$

$$
\leq \frac{1}{\sqrt{\delta}(n+1)} \sum_{i=1}^{n} \mathbb{P}\left[\left|\hat{\mu}_{\varphi\backslash i}(X_{n+1}) - \hat{\mu}^*_{\varphi\backslash i}(X_{n+1})\right| + \left|\hat{\mu}_{\varphi\backslash i}(X_i) - \hat{\mu}^*_{\varphi\backslash i}(X_i)\right| > 2\varepsilon\right]
$$

$$
\leq \frac{2n}{\sqrt{\delta}(n+1)} \mathbb{P}\left[\left|\hat{\mu}_{\varphi\backslash n}(X_{n+1}) - \hat{\mu}^*_{\varphi\backslash n}(X_{n+1})\right| > \varepsilon\right].
$$

The second inequality is the Markov's inequality, and the last step uses the exchangeability of the data points. Plugging in (S9),

$$
\mathbb{P}\left[\hat{C}^{(2\varepsilon+2\varepsilon')\text{-J+aB}}_{\alpha,n,B}(X_{n+1}) \not\supseteq \hat{C}^{*2\varepsilon'\text{-J+}}_{\alpha',n}(X_{n+1})\right] \leq 2\sqrt{\delta},
$$

implying (S10). This completes the proofs for Theorems S1 and S2. $\qquad\square$

## C  Jackknife-minmax-after-bootstrap

As in Barber et al. [1], we may also consider the *jackknife-minmax-after-bootstrap*, which constructs the interval

$$
\hat{C}^{\text{J-mm-aB}}_{\alpha,n,B}(x) = \left[\min_i \hat{\mu}_{\varphi\backslash i}(x) - q^-_{\alpha,n}\{R_i\}, \ \max_i \hat{\mu}_{\varphi\backslash i}(x) + q^+_{\alpha,n}\{R_i\}\right].
$$

The original jackknife-minmax satisfies $1-\alpha$ lower bound on the coverage, and the same modification of the jackknife+ proof is applicable here, ensuring a $1-\alpha$ lower bound on the coverage of the jackknife-minmax-after-bootstrap with the same caveat of a random $B$. However, as for the non-ensembled version, the method is too conservative, and is not recommended for practice.

## D  Supplementary experiments

In this section, we report the results on additional experiments.

First, we provide a more detailed description of the ensembles and the jackknife-type constructions considered. Let $\mathcal{R}_{\varphi,B}$ denote the ensemble Algorithm 1 that first generates $B$ bootstrap replicates of the given training data set, calls on a base regression method $\mathcal{R}$ to fit a model to each generated data set, after which the results are combined through an aggregation function $\varphi$. For $\mathcal{R}$, we use one of RIDGE, RF, or NN, which are described in Section 5. For $\varphi$, we use one of MEAN, MEDIAN, or TRIMMED MEAN:

- MEAN is the arithmetic mean, i.e., $\varphi(y_1, \dots, y_k) = k^{-1}\sum_{i=1}^{k} y_k$.

- MEDIAN is the middle value of the given list, i.e., for odd $k$, $\varphi(y_1, \dots, y_k)$ is the $(k+1)/2$-th smallest number of the list $\{y_1, \dots, y_k\}$; for even $k$, it is the average of the $k/2$-th and the $(k+2)/2$-th smallest.

- TRIMMED MEAN is the arithmetic mean of the middle 50% of the given list, i.e., $\varphi(y_1, \dots, y_k) = (\lceil 0.75k\rceil - \lfloor 0.25k\rfloor)^{-1}\sum_{i=\lfloor 0.25k\rfloor+1}^{\lceil 0.75k\rceil} y_{(i)}$, where $y_{(1)} \leq \cdots \leq y_{(k)}$ is the sorted list. We use `scipy.stats.trim_mean` with `proportioncut=0.25`.

J+AB is defined in Algorithm 2. J+ ENSEMBLE refers to the following application of the jackknife+ [1] with the ensemble learner $\mathcal{R}_{\varphi,B'}$ (with hyperparameter $B'$):

---
**Algorithm 1** J+ ENSEMBLE
---
**for** $i = 1, \ldots, n$ **do**

    Compute $\hat{\mu}_{\backslash i}^{\text{J+ ENSEMBLE}} = \mathcal{R}_{\varphi, B'}(\{(X_j, Y_j)\}_{j=1, j \neq i}^n)$

    Compute the residual, $R_i^{\text{J+ ENSEMBLE}} = |Y_i - \hat{\mu}_{\backslash i}^{\text{J+ ENSEMBLE}}(X_i)|$.

**end for**

Compute the ensembled prediction interval: at each $x \in \mathbb{R}$,

$$\hat{C}_{\alpha, n, B'}^{\text{J+ ENSEMBLE}}(x)$$
$$= \left[ q_{\alpha, n}^- \{\hat{\mu}_{\backslash i}^{\text{J+ ENSEMBLE}}(x) - R_i^{\text{J+ ENSEMBLE}}\}, q_{\alpha, n}^+ \{\hat{\mu}_{\backslash i}^{\text{J+ ENSEMBLE}}(x) + R_i^{\text{J+ ENSEMBLE}}\} \right].$$

---

By contrast, J+ NON-ENSEMBLE applies the jackknife+ around the *base* learning algorithm $\mathcal{R}$:

---
**Algorithm 2** J+ NON-ENSEMBLE
---
**for** $i = 1, \ldots, n$ **do**

    Compute $\hat{\mu}_{\backslash i}^{\text{J+ NON-ENSEMBLE}} = \mathcal{R}(\{(X_j, Y_j)\}_{j=1, j \neq i}^n)$

    Compute the residual, $R_i^{\text{J+ NON-ENSEMBLE}} = |Y_i - \hat{\mu}_{\backslash i}^{\text{J+ NON-ENSEMBLE}}(X_i)|$.

**end for**

Compute the *non-ensembled* prediction interval: at each $x \in \mathbb{R}$,

$$\hat{C}_{\alpha, n}^{\text{J+ NON-ENSEMBLE}}(x)$$
$$= \left[ q_{\alpha, n}^- \{\hat{\mu}_{\backslash i}^{\text{J+ NON-ENSEMBLE}}(x) - R_i^{\text{J+ NON-ENSEMBLE}}\}, q_{\alpha, n}^+ \{\hat{\mu}_{\backslash i}^{\text{J+ NON-ENSEMBLE}}(x) + R_i^{\text{J+ NON-ENSEMBLE}}\} \right].$$

---

Note that for J+AB, we match the *expected* number of models aggregated in each leave-one-out model $\hat{\mu}_{\varphi \backslash i}$ to $B'$, the fixed number of models aggregated in each ensembled model $\hat{\mu}_{\backslash i}^{\text{J+ ENSEMBLE}}$ by drawing $B \sim \text{Binomial}(\tilde{B}, (1 - \frac{1}{n+1})^m)$ with $\tilde{B} = [B'/\{(1 - \frac{1}{n+1})^m (1 - \frac{1}{n})^m\}]$, where $[\cdot]$ refers to the integer part of the argument.

In Section 5, we compared J+AB with J+ ENSEMBLE and J+ NON-ENSEMBLE for $\mathcal{R} = \text{RF}$ and $\varphi = \text{MEAN}$ using MEPS data set. Here, we use all nine combinations of $\mathcal{R}$ and $\varphi$, and also expand the number of trials to ten. The results are presented in Figures S1 and S2. In Table S1, we report the average wall-clock time for each $\mathcal{R}$-$\varphi$ combination for $m = 0.6n$. The additional results lend extra support to the conclusion that the J+AB is a computationally efficient alternative to J+ ENSEMBLE, which yields more precise confidence intervals than J+ NON-ENSEMBLE when ensembling improves the precision of the base regression method.

Table S1: Average wall-clock times in seconds over 10 independent splits of MEPS data set. ($m = 0.6n$ and sampling with replacement)

| | ENSEMBLE | | | |
|---|---|---|---|---|
| $\mathcal{R}$ | $\varphi$ | J+AB | J+ ENSEMBLE | J+ NON-ENSEMBLE |
| RIDGE | MEAN | 0.22 | 1.93 | 0.40 |
| | MEDIAN | 0.59 | 2.71 | |
| | TRIMMED MEAN | 0.57 | 2.58 | |
| RF | MEAN | 6.03 | 43.87 | 4.60 |
| | MEDIAN | 6.24 | 44.29 | |
| | TRIMMED MEAN | 6.84 | 39.23 | |
| NN | MEAN | 16.82 | 160.00 | 14.12 |
| | MEDIAN | 17.80 | 169.31 | |
| | TRIMMED MEAN | 17.48 | 162.84 | |

The other experiment in Section 5 checked the coverage of the J+aB method using a Binomial $B$. Only the results for $m = 0.6n$ on the MEPS data set were reported. Here, we present the complete set of results for $m/n = 0.1, 0.2, \ldots, 1.0$ and for all three data sets. We expand the number of trials to ten, and increase $n = 200$ and the hyperparameter $\tilde{B}$ in $B \sim \text{Binomial}(\tilde{B}, (1 - \frac{1}{n+1})^m)$. In addition, we report the results with $B'$ FIXED (as opposed to RANDOM). For J+AB FIXED, we fix the total number of bootstrap replicates at $B' = 50$. For J+AB RANDOM, we set the *expected* total number of bootstrap replicates to $B'$ by taking $\tilde{B} = [B'/(1 - \frac{1}{n+1})^m]$. Consistent with what we saw in Table 1, the coverage respects the $1 - 2\alpha$ lower bound, and in fact, stays close to $1 - \alpha$ in all considered scenarios. In addition, Figures S3–S8 show that J+AB RANDOM and J+AB FIXED have essentially the same behavior in practice.

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

Figure S1: Comparing the coverage of J+AB, J+ ENSEMBLE, and J+ NON-ENSEMBLE for varying $m/n$ on MEPS data set. The solid lines show the average, and the shaded areas show $+/-$ one standard error over 10 independent splits of the data. The black dash-dotted line marks the $1 - \alpha$ target coverage level.

Figure S2: Comparing the interval width of J+AB, J+ ENSEMBLE, and J+ NON-ENSEMBLE for varying $m/n$ on MEPS data set. The solid lines show the average and the shaded areas show $+/-$ one standard error over 10 independent splits of the data.

# Coverage on COMMUNITIES

Figure S3: Average coverage of J+AB RANDOM and J+AB FIXED on COMMUNITIES data set. The solid lines show the average and the shaded areas show $+/-$ one standard error over 10 independent splits of the data. The black dash-dotted line marks the $1 - \alpha$ target coverage level.

**Coverage on BLOG**

Figure S4: Average coverage of J+AB RANDOM and J+AB FIXED on BLOG data set. The solid lines show the average, and the shaded areas show $+/-$ one standard error over 10 independent splits of the data. The black dash-dotted line marks the $1 - \alpha$ target coverage level.

**Coverage on MEPS**

Figure S5: Average coverage of J+AB RANDOM and J+AB FIXED on MEPS data set. The solid lines show the average, and the shaded areas show $+/-$ one standard error over 10 independent splits of the data. The black dash-dotted line marks the $1 - \alpha$ target coverage level.

Figure S6: Average interval width of J+AB RANDOM and J+AB FIXED on COMMUNITIES data set. The solid lines show the average, and the shaded areas show $+/-$ one standard error over 10 independent splits of the data.

Figure S7: Average interval width of J+aB RANDOM and J+aB FIXED on BLOG data set. The solid lines show the average, and the shaded areas show $+/-$ one standard error over 10 independent splits of the data.

Figure S8: Average interval width of J+AB RANDOM and J+AB FIXED on MEPS data set. The solid lines show the average, and the shaded areas show $+/-$ one standard error over 10 independent splits of the data.