[Reviews · NeurIPS 2020]

Review 1

Summary and Contributions: - This paper proposes a method which integrates ensemble learning with the recently proposed jackknife+, called jackknife+-after-bootstrap(J+aB). - The method is implemented by tweaking how the ensemble aggregates the learned predictions. - The paper proves that the coverage of a J+aB interval is at worst 1 − 2\alpha for the assumption-free theory. - They provide a cost-free (in terms of model fitting) solution for constructing predictive intervals making use of only the bootstrapped and their fitted models. - In summary, the proposed method offers confidence intervals in an assumption-free setting, through a computationally efficient wrapper method for ensemble learning.

Strengths: - Extention of previous works to consider confidence intervals - Strong theoretical justification of the methods. Explained and justified the design choices theoretically. - One negative aspect of ensembles is the computational complexity, this method reduces the computational complexity compared to ensembles. Reducing the number of calls to the base regression method, making it similar to a single ensemble. - Even for the setting, where the computational complexity of the model evaluation/aggregation remains low, the J+aB algorithm is able to provide confidence intervals at no extra cost.

Weaknesses: Major ----- - Experiment section needs quite some work. The quality of the experiment section and the experiments do not seem to match that of the rest of the paper. In addition, there is a lack of experiments showcasing the benefits of the method discussed earlier. A stronger experiment setup would strengthen the paper and better support the theory/claims from the rest of the paper. - I find it hard to understand the results in Tab. 1. It is unclear, to me, what is the take-away message in this table. There is a lack of a proper discussion regarding this result. It would make it easier to understand the experiment if more discussion around the result is provided. - Based on the results provided in the appendix the paper lacks the comparison of the coverage and width measures to the two baseline methods for the other two datasets COMMUNITIES and BLOG. Based on the provided result on MEPS on other base regression methods (RF, NN, Ridge), it is clear that results (RF regressor) shown in the main paper is an exception where J+aB performs better than the baseline methods. J+aB performs almost the worst for the Ridge regressor and very similar to the baseline methods for the NN regressor. Even though for the second case where J+aB performs similar to J+ensemble, it is still better as it significantly reduces the computation time, though for the Ridge case it is consistently out-performed by J+non-ensemble where the computational gains are less. This observation is similar for Coverage and Width. - Fig. 1 is missing axis labels and font size is very tiny. Usually, such a point would fall under "minor" weaknesses, but as it is the only figure/experiment comparing to the baseline here it falls under "major" as it is an important figure. As there quite some space left blank in the figure, I would suggest that this figure needs large fonts and axis labels. Minor ---- - Tab. 1: best performing method could have been bold.

Correctness: Yes partially. The weaker experiment section makes it hard to judge if the empirical methodology is correct.

Clarity: Yes. Up until the experiment section, it is well written. The experiment section lacks discussion.

Relation to Prior Work: Yes, the difference is clearly discussed.

Reproducibility: Yes

Additional Feedback: -Strengthening the experiments would greatly help the paper. -Improve the quality of the figures ------------------- Review Update ----------------- I thank the authors for their response. I still think the papers experimental section needs quite some changes to improve the paper. The experimental section has much lower quality (something I feel is hard to change through a 1-page author response) than the rest of the paper, therefore failing to showcase the benefit of obtaining the theoretical guarantees (i.e. the heavy focus of the current version of the paper). I would encourage the authors to better showcase the benefit of their method to better support the theory. Already discussing/explaining some of the weaker results shown in the appendix could be a good starting point.


Review 2

Summary and Contributions: This paper considers generating prediction intervals for data drawn from an iid distribution. It is model agnostic, operating on the residuals of fitted models. Specifically, the proposal modifies the recent jackknife+ algorithm, which has provable coverage of 1-2\alpha. A new version, J+aB, is proposed that works more efficiently with ensemble models, yet maintains the same 1-2\alpha coverage guarantee.

Strengths: The paper presents a solid set up and background to the proposal. It’s maturely written, with good attention to detail. The proposal seems valuable. Applying the jackknife+ naively to an ensemble of B models on n data points requires training of O(nB) models. J+aB instead requires training of only O(B) models, and yet is proven to have the same coverage guarantee of the naïve procedure.

Weaknesses: The experimental section is the weakest part of the paper. The models, datasets, and compute are very small. The reported metrics and analysis don’t offer much insight. Running experiments more similar to [1] might have been helpful. 1) It was stated that n=40, m=0.6n, E[B]=20. Does this mean we end up with (around) 20 models, each trained on 24 data points? This seems very small relative to the size of the full dataset. 2) “J+ ENSEMBLE took about 30 to 60 times longer to finish running compared to the two other methods” – disagrees with table S1, which shows ~10x increase. 3) I expected Figure 1 to match with Figure S1,S2 (RF, mean) but there seem to be differences? 4) The described NN implementation is rather outdated. I’d have loved to see whether improvements over more ad-hoc NN ensembling methods, e.g. [2], could be demonstrated. There are certain areas of the theory which I don’t fully follow/don’t make intuitive sense to me. 1) I’m surprised there are no further requirements on \tilde{B} and m in theorem 1. Would the procedure be valid if they were both set to 1 or 2? 2) I don’t understand the intuition for why B must be a random variable. It feels like a contrived condition to make the proof work. I can’t see any difference (and am not sure I would expect to) in appendix plots S3 onwards comparing random vs fixed B. 3) Would be useful to summarise complexity for each method Jackknife/J+/J+aB, both in terms of model training and prediction, for single models and ensembles where appropriate. (I would like to see this in the rebuttal.) Overall, I have the feeling J+aB has been carefully designed to respect the theoretical coverage guarantee of J+, i.e. 1-2\alpha, which feels opposite to the usual development path of algorithms. Something must be lost by using less models compared to the naïve J+non-ensemble, though this isn’t discussed. Ultimately the theoretical guarantee is much looser than what both these and other methods seem to produce in practise. I therefore see the merit of the work being more theoretical than practical. A personal view on the topic: Although iid data is a common assumption in ML, many ML communities seems to be growing more interested in methods addressing out-of-distribution/distribution drift etc. [1] Adaptive, Distribution-Free Prediction Intervals for Deep Networks [2] Simple and Scalable Predictive Uncertainty Estimation using Deep Ensembles

Correctness: The abstract’s claim of ‘no assumptions on the distribution of the data’ could mislead - the iid/exchangeable assumption should be acknowledged as it is in the intro. I think some discussion should be added describing the increased complexity when evaluation is not negligible - around line 171: “where both model evaluation and aggregation remain relatively cheap, our J+aB algorithm is able to output a more informative confidence interval at virtually no extra cost beyond what is already necessary to produce a single point prediction with just the ensemble method”.

Clarity: The paper is well written with good polish, both in language and math. It’s well structured. The term ‘inference’ is used in the title and abstract without being defined. I’d recommend briefly describing it in the context of this paper, since it’s an overloaded term.

Relation to Prior Work: Yes.

Reproducibility: Yes

Additional Feedback: Figure 1 has no x axis label – I assumed it followed from the appendix plots. ____________________ Post-rebuttal comment Thank you for your rebuttal, and in particular the breakdown of computational complexity on line 32. I haven’t felt motivated to revise my score – I remain of the opinion that the theory in the paper is valuable. If there is to be a further iteration of the paper, I’d encourage investigating and discussing the practical implications of the modifications, as also suggested by R5.


Review 3

Summary and Contributions: The authors propose the J+aB ("Jackknife+ after bootstrap") approach to build predictive confidence intervals of an ensemble for new observations. This approach doesn't require training additional models; it directly uses the models that constitute the ensemble and which are fitted on the bootstrapped samples. As a result of that, building the interval can be done faster and at a lower computational cost than other approaches (e.g. Jackknife+). The authors provide a non-asymptomatic distribution-free predictive coverage guarantee.

Strengths: This paper builds on the Jackknife+ approach. It extends the approach to efficiently compute predictive confidence intervals for ensembles. It also provides a non-asymptomatic distribution-free predictive coverage guarantee. Being able to efficiently build predictive intervals is important as the demand for measuring uncertainty related to models used in certain application domains is increasing.

Weaknesses: Looking at algorithm 2, I wonder if the speed up observed when building the predictive confidence interval can sometimes come at the cost of a larger predictive interval. Given that the residual R_i is computed by applying the aggregation function only to the models whose training set didn't include X_i, can we get into a situation where the number of such models becomes small (compared to B), therefore limiting the generalization ability (or smoothing ability) of the aggregation function, which would lead to a higher residual? This effect might be attenuated by the fact that the predictive interval depends on quantiles and that not all residuals might be large. But it would be important to understand how the width of the predictive interval evolves as a function of B, m, n, and possibly R. Looking at figures S2, S6, and S7, we see that the width of the predictive interval sometimes increases for some models (mostly NN and RF) as m --> n. Assuming that my concerns are valid, can the authors clarify the trade offs between the speed up of j+aB and the width of the predictive intervals that it generates? In their rebuttal, the authors pointed out that the number of models in the J+AB approach needs to be drawn from a binomial distribution which depends on the number of models in J+Ensemble and which also depends on n and m. But it's still not clear to me whether J+AB and J+Ensemble lead to predictive intervals of the same width. The authors stated in their rebuttal that this is shown in the experimental results. But they only present results for one dataset. No results comparing J+AB and J+Ensemble are provided for the other 2 datasets. And I don't see a discussion of the width of the predictive intervals in the theoretical part of the paper.

Correctness: I have not verified the correctness of the proof of Theorem 1 which provides the distribution-free predictive coverage guarantee. As for the speed up (or lower computational cost) claims made by the authors (compared to the "j+ ensemble" approach), they are simple to verify from Algorithm 2 and are supported by the empirical evaluation. The claims made in the paper do not describe the impact of the approach on the width of the predictive intervals. I think that this topic needs to be discussed. The empirical evaluation needs to be improved by presenting the coverage and width figures for "j+ aB", "j+ ensemble" and "j+ non-ensemble" for all 3 datasets, instead of only providing them for MEPS. It would also be important to explain how the width of the predictive interval evolves as a function of B, m, n, and R. I think that the residual at line 74 should use \mu{hat} and not \mu{hat} excluding i.

Clarity: The paper is very well written. The structure is well defined. The text is clear and easy to follow. And the notations are clear.

Relation to Prior Work: The authors describe prior work and briefly explain how their work is different. I am not familiar with the literature in this area to confirm that all important and relevant prior work has been documented.

Reproducibility: Yes

Additional Feedback: In order to improve reproducibility and given that the authors use the default settings of methods implemented in scikit-learn, it would be important to either document the version of scikit-learn that was used or to document the values of the default settings.

[Author Response · NeurIPS 2020]

We thank the reviewers for their feedback. Below we respond to some of the main concerns.

**Clarification of the experimental goals** R1 and R3 were dissatisfied with the small scale of our experiments. We are
happy to run any additional experiments that are deemed crucial for better understanding of our method. In fact, we are
happy to leave the choice of additional data and ensemble methods up to the reviewers. However, we would like to
first clarify what we were hoping to illustrate with the current setup, and further discuss what constitutes meaningful
comparisons for a wrapper method such as our J+AB that comes with a model-free guarantee.

The primary goal of our experiments is to demonstrate that our method achieves near $1 - \alpha$ coverage numerically
(according to the theory, $1 - 2\alpha$ is guaranteed). The secondary goal is to verify that although J+AB runs faster than
J+ ENSEMBLE, $\hat{C}^{\text{J+AB}}_{\alpha,n,B} \approx \hat{C}^{\text{J+ ENSEMBLE}}_{\alpha,n,B'}$. The final, lesser goal is to relate known stabilizing properties of bagging by
comparing J+AB (or J+ ENSEMBLE) vs J+ NON-ENSEMBLE. These goals are either stated or implied in Lines 251-6,
294-303, but we promise to make them more explicit in the camera-ready version.

There were two main reasons for running experiments on a small scale with a couple of data sets. The first is the page
limit. The second is the cost of running J+ ENSEMBLE; we needed the experimental parameters to be quite small to be
certain of obtaining results by the deadline. This is remarked in Lines 284-6. However, given our experimental goals,
we did not see the lack of scale as a significant defect. The advantage of model-free framework is that our coverage
guarantee is impossible to break irrespective of the data and the choice of ensemble. We can always choose to look
at more data sets and more ensemble architectures, but this will only produce more plots that all look very similar.
Meanwhile, the secondary goal amounts to a sanity check, and we have said that the final goal is of lesser significance.

The issue of width is certainly of interest, but here, we would argue that the only meaningful comparisons are those with
other wrapper methods. For example, a split conformal variant is competitive with J+AB in terms of computational
cost, but is expected to lose in terms of statistical efficiency. Although this is rather obvious by construction, it may be
interesting to investigate whether this would translate to meaningful differences in performance. This is a comparison
of efficiency that we are happy to add to our current results. Otherwise, the precision of the intervals would be most
heavily affected by the fit of the chosen ensemble with the data. However, as we have ceded this choice to the user,
opting to develop a fully flexible method that works irrespective of the quality of this choice, we believe that additional
comparisons involving more particular instances of $\mathcal{R}$ or $\varphi$ are not as useful and tangential to the topic.

**Breakdown of computational complexity** R3 requested a summary of computational complexity. Here, we provide
the total number of occurrences for three different types of operations, which can be used to derive the final cost. We
focus on bootstrapping, and match the number of models as in Supplement, Lines 166-9. The table below demonstrates
that if the model-fitting cost dominates, the cost of J+AB is roughly that of obtaining a single ensemble prediction. We
do not claim any advantage for our method when the cost is dominated by aggregation or evaluation. See Line 170.

|  | #calls to $\mathcal{R}$ | #evaluations |
|---|---|---|
| JACKKNIFE | $n+1$ | $n+1$ |
| JACKKNIFE+ | $n$ | $2n$ |

|  | #calls to $\mathcal{R}$ | #calls to $\varphi$ | #evaluations |
|---|---|---|---|
| J+ ENSEMBLE | $B'n$ | $n$ | $2B'n$ |
| J+AB (on average) | $B'/(1-\frac{1}{n})^m$ | $n$ | $2B'/(1-\frac{1}{n})^m$ |
| ENSEMBLE | $B'$ | $1$ | $B'$ |

**Tradeoff between computational and statistical efficiency** R5, as well as R3, expressed concerns about a tradeoff
between computational and statistical efficiency for the J+AB vs J+ ENSEMBLE comparison. The short answer is that
one method does not always win over the other. See Figure S2. First, since $\tilde{B}$ is a user-specified parameter, it can be
picked so that the numbers of models in $\hat{\mu}_{\varphi \setminus i}$ are matched on average. See Supplement, Lines 166-9. Second, the more
important difference is the *correlation* among $\{\hat{\mu}_{\varphi \setminus i}\}_{i=1}^{n}$. Conditional on the observed data, $\{\hat{\mu}_{\varphi \setminus i}\}_{i=1}^{n}$ are dependent
in the case of J+AB and independent in the case of J+ ENSEMBLE. (Note that unconditionally they are always highly
correlated for both.) What this means for the precision is expected to depend on the data and the choice of ensemble. In
any case, this difference is expected to be much smaller than, say, that for the J+AB vs split conformal comparison.

**R1** 2) The takeaway of Table 1 is in Lines 289-91 (as well as Lines 251-2). It is not our goal to see which among the
nine (all *instances* of J+AB) performs best. 3) The results for RF vs RIDGE are completely expected given the known
results on bagging. The point we are trying to illustrate is in Lines 295-7. 4) Figure 1 shall be amended.

**R3** 3) The number of trials was *doubled* for Supplement C, which reduced the standard errors. 4) J+AB can be applied
to any ensemble algorithm in the form of Algorithm 1 as long as it is agnostic to the ordering of the input data. $\mathcal{R}$ or $\varphi$
may be arbitrarily complicated, e.g., $\mathcal{R}$ may involve built-in hyperparameter tuning; $\varphi$ may have adaptive weights. 1)
No further condition on $\tilde{B}$ is necessary. The validity comes from construction of an exchangeable array of residuals. 2)
For fully distribution-free guarantee, a random $B$ is necessary, as the array is not exchangeable with $B$ fixed. Figure S3
assumes *bagging*, so fixes a particular resampling and a particular $\varphi$. Also, the title and the abstract will be amended.

[Meta-Review · NeurIPS 2020]

This submission is a hard one. Given the reviewer scores and their relatively low confidence in their write-ups I reviewed this paper myself. (The reviewers all discussed their comments and updated their reviews as well.) I am torn. On one hand, the reviewers raise valid concerns about the experimental study; the results in the appendix highlight important aspects of their method, as applied to non-tree based methods. These are not addressed in the main text. And the authors' response to this point is very unsatisfying. ("Not enough space" and "our theory proves everything" are not valid responses to what can come across as cherry-picked results.) A trival response would have been "yes, we can shorten the unnecessarily verbose section 2.2 and bring these results into the main text -- and here's how to interpret these graphs". Instead, the authors choose to defend a 1 page Section 2.2, which constitutes basic background for anyone interested in reading this paper in the first place. On the other hand, the method is clever. It can save significant computation and the theory appears correct. In cases where ensemble learning is used, this approach seems like a practical method. In spite of my frustration with the authors' response, I am recommending that this paper is accepted. However, I will be very disappointed if there isn't a significant effort made into improving the experimental study and shifting of key experimental results from the appendix into the main text, especially with the extra page that accepted will enjoy this year.